# Nice for Whom? A Dangerous, Not-So-Nice, Critical Race Love Letter

G. T. Reyes

Department of Educational Leadership, California State University East Bay, 25800 Carlos Bee Blvd., Hayward, CA 94542, USA; g.reyes@csueastbay.edu

**Abstract:** In this article, I critically analyze and respond to empirical data in the form of racialized discourse—specifically, racist messages sent directly to me as a result of my previously published article entitled, "A Love Letter to Educational Leaders of Color: CREWing UP with Critical Whiteness Studies". Being informed by a robust racial analysis of acts that reinforce white supremacy, this article will likely be perceived as *not* nice by those who benefit from and work to protect white supremacy. Likely, *I* will be the one accused of being hateful, divisive, and even racist. In order to interrogate the weaponization of this conception of "niceness", my analysis will be driven by Critical Race Hermeneutics with white emotionality and whitelashing used as interpretive lenses. As this article's engagement with these critical race frameworks poses a threat to those who benefit from racism, this is a dangerous, not-so-nice critical race love letter.

**Keywords:** critical race theory; critical whiteness studies; educational leadership

## 1. Introduction: Dear Beloved Community

I write to you with an intention of expressing radical, decolonial, and humanizing love, because frankly, we all need it. And I must be clear about what I mean by love, because oftentimes, it is believed to be an irrational or romanticized feeling and emotion that has no place in our work. Rather, I speak of love as intention, verb, prayer, and praxis [1]. I speak of love that unapologetically asserts our full humanity and wellbeing [2]. I speak of love that cultivates trusting, accountable, and interdependent relationships [3]. I speak of the kind of love that attends to our human need to feel loved, to share love, to sustain joy, and to heal from systems of oppression that inflict harm and work to sustain woundedness [4]. Such love is grounded in an analysis of colonialism and racism, as well as a profound purpose that drives radical acts to improve not only one's own conditions but also those of others [5]. I speak of the kind of love that feeds the work to reroot, restore, rethink, and reimagine our world [6]. But why do such a thing? Why write to express love, particularly in a scholarly research journal?

First, I express love by specifically writing in the form of a critical race love letter. In doing so, no matter who you are and what your positionality is, I hope that you receive my offering with grace, openness, and self-reflection. I have found that using the genre of letter writing allows me to cultivate greater intimacy with you as the reader and therefore, relationality and solidarity. For transparency, I write in the form of a critical race love letter to not only recognize but to also practice the tenets of critical race theory (CRT) as it relates to education [7,8]. Though it is not my focus to review the CRT tenets in detail since they have been widely and thoroughly presented and discussed elsewhere [7–9], it is important for me to make clear that this love letter starts from the place that race and racism not only exist but are pervasive, saturating, and a normalized part of our U.S. society. By "normalized", I do not mean to say that racism should feel "normal". I use the verb form—normalize—to indicate that it is a structured process. With this recognition, this love letter does not simply theorize for the sake of theorizing. This love letter is grounded

and centered upon experiential knowledge, particularly from those in education like me who are BIPOC and actively work to challenge racism. As such, this love letter draws from that experiential knowledge while drawing from critical whiteness studies and critical race theory to challenge the mundane, prosaic ways that whiteness functions within a larger system that works to protect and defend itself. In doing so, this love letter seeks to empower BIPOC educational leaders with a method to challenge racialized discourse.

As such, this letter is not so nice. Similarly, the song *Nice for What* [10], performed by music artist Drake, has intentions of not-so-niceness as well. In his song, he presents both an affirmation of women as well as a critique of the inequities they experience. He highlights the resilience and strength of women in spite of the numerous social, financial, and professional inequities that they face. Oftentimes, critiquing oppressive systems is perceived by those who benefit from those systems as not being nice. In the case of women, their perceived not-so-niceness is further demonized by name calling (e.g., being called a bitch) or being condescendingly told to change their behavior by doing pleasant things like "smiling more". As such, Drake expresses that women do not need to be "nice" for the sake of a man or broader yet, the institutions that perpetuate patriarchy, so he asks the question, nice for what? Similar to the challenge embedded in *Nice for What*, this letter adds the question, *Nice for Whom?* Specifically, this letter challenges the expectations to be "nice" when critiquing white supremacy even within a pervasive white supremacist context that will not only reject the presence of white supremacy but will fight tooth and nail to protect it. As such, this letter has no choice but to be not so nice to those who feel most threatened by racial justice.

By "not-so-nice", I do not mean that my letter is hateful or seeks to harm. The subjective and socially constructed nature of niceness often invisibilizes its underlying dominant ideologies [11] as well as its dual nature. For instance, if I am informed by a robust racial analysis of acts that reinforce white supremacy, my words will likely be perceived as *not* nice by those who benefit from and work to protect white supremacy. Likely, *I* will be the one accused of being hateful, divisive, and even racist. Conversely, if a white person says something that I might interpret as racist, and if that white person believes they had good intentions or was attempting to be nice in whatever they were saying, then the impact on me does not matter. I took the person the wrong way, I was being too sensitive, or I was not able to take a joke. Castagno helps us to understand this dynamic by adding, "having good intentions is a critical component of Niceness. In fact, as long as one means well, the actual impact of one's behavior, discourse, or action is often meaningless" [11] (p. x). She further elaborates,

> Within a frame of Niceness, oppressive actions are not actually oppressive; they are just hurtful. They are therefore assumed to be the result of individuals who have made bad choices or who just do not know any better. This framing diverts attention away from patterned inequity, structural oppression, and institutional dominance [11] (pp. xi–xii).

So then, let me be explicit and transparent about one purpose of this letter. The not-so-niceness of this critical race love letter is driven by a radical, decolonial, and humanizing love applied to a very specific situation that I experienced. Within this letter, I critically analyze and respond to empirical data in the form of discourse—specifically, racist messages sent directly to me as a result of my previously published article entitled, "A Love Letter to Educational Leaders of Color: CREWing UP with Critical Whiteness Studies" [6]. My analysis will be driven by Critical Race Hermeneutics [12] with white emotionality [13,14] and whitelashing [15] used as interpretive lenses. It is my hope then that in this critical race love letter that you presently read, you will be affirmed in your own anti-racist work where you have also likely received racist messages at a minimum, if not more overt assaults such as death threats or the threat of being fired from your institution. As such, through this letter, I wish you to know that you are not alone, and that we, particularly as BIPOC, have to work hard to love ourselves, particularly within a country that does not love us back and would rather us hate ourselves [14].

## 2. Background: The Danger of CRT and the Need to Clap Back for Self-Preservation

To love ourselves, particularly within a country that does not love us back and would rather us hate ourselves necessitates a commitment to journey through a long-haul process of making critical sense of our experiences, values, beliefs, history, and world. As I have mentioned earlier, this critical race love letter works in praxis while engaging CRT tenets. Though it is not my focus to review the CRT tenets in detail since they have been widely and thoroughly presented and discussed elsewhere [7–9], I do feel responsible in identifying how my love letter activates those tenets. Specifically, in the next paragraph, I indicate where my letter activates a CRT tenet with bolded lettering.

CRT began in the late 1970s from critical legal scholars of color as a response to be able to analyze and respond to the ways that the law was not only unable to end racism and racial inequality but also perpetuated it [9,16]. It is important that I highlight this origin from critical legal studies—even within a critical race love letter—so as to make clear that I **begin from the place that acknowledges race and racism** as pervasive, saturating, and sustaining in the United States, yet systemically invisibilized in institutional and cultural ways. Because of this, we (albeit very differently as a process and as a consequence, both BIPOC and white people experience race and racism) not only face material consequences, but we also experience emotional, mental, and psychic ones. As such, this letter employs **counter-storytelling** that centers the experiences of BIPOC as legitimized truth. It does so by drawing from a **variety of discourses** to **interrogate normalized racist behaviors and discourse**. Through such analyses, we become well informed to **counteract racism by engaging in reflexive action** that includes the healing that must be done from experiencing racist systems that work to sustain our wounds.

This engagement with CRT, however, poses a threat to those who benefit from racism. So, in order to love ourselves, particularly within a country that does not love us back and would rather us hate ourselves, we must be critically aware that the anti-racist work we do is, has been, and will continue to be dangerous. There are no shortage of receipts in history to illustrate this danger. In the 1930s, for instance, New Haven residents passive-aggressively threatened the creation of a Black college in the area by saying it would be an "unwarranted and dangerous undertaking" [17] (p. 12). (Givens, 2021, p. 12). Threats such as this did not go unnoticed as systemic. We are reminded of this by Carter G. Woodson, who wrote *Miseducation of the Negro* [18] (1990) in the 1930s when he illuminated that the institution of schooling positions us on "dangerous ground" (p. 24) because of the systemic way that the "inferiority of the negro is drilled into him in almost every class he enters and almost every book he studies" (p. 2). He provoked us to ask: what are we being taught, not taught, why, and towards what aims? Asking such questions pose a risk, particularly from those committed to protecting curriculum that maintains the status quo. James Baldwin extended the interrogation of this risk in his 1963 *A Talk to Teachers* [19]. In that work, he asserted that teachers must understand the risks of our "dangerous times" (p. 325) because we have generations of systematized miseducation to counteract within a society that does not fundamentally want to change. As educators and educational leaders take on these risks, they must then be reminded of the political purpose in doing so. They must know that they are doing what Gloria Ladson-Billings called "dangerous work" [20] (p. 240), which destabilizes mainstream ideologies, norms, behaviors, attitudes, practices, and systems that dominant groups attempt to protect at all costs.

If we are to maintain our humanity, dangerous work is necessary. Remaining steadfast in such work and doing it with integrity, grace, and vigilance requires us to love ourselves. The assertion of our love requires us to challenge that which attempts to make us hate ourselves. Thus, we must adopt and adapt race conscious frameworks to empower how we critique whiteness and systemic antiblackness, call out racism, work towards racial equality and equity, and overall challenge white supremacy. In spite of the danger of such work, we still do it. To love ourselves, particularly within a country that does not love us back and would rather us hate ourselves, then requires persistent engagement when the assertion of our love is met with an opposing aggressive and reactionary behavior.

Throughout this journey Ladson-Billings [21] (1998) reminds us, "adopting and adapting CRT as a framework for educational equity means that we will have to expose racism in education and propose radical solutions for addressing it. We will have to take bold and sometimes unpopular positions. We may be pilloried figuratively or, at least, vilified for these stands" (p. 22).

There is no shortage of receipts throughout history that demonstrate public vilification for taking stands against racism and white supremacy. Rather than tracing this history to its colonial roots [22], I will only focus on illustrating the more recent assaults against CRT. Though CRT has been applied by education scholars in research for approximately three decades, attacks against it significantly surged following the murders of Breonna Taylor in March 2020 and George Floyd in May 2020, which occurred early during the global pandemic [23,24]. The nature of the global pandemic and its shelter-in-place orders created a social microscope on a variety of racial, health, economic, and political inequities that were not new but rather amplified as a result of the heightened and more focused attention. For instance, the Black Lives Matter (BLM) movement, which had existed for approximately seven years by then, significantly expanded to become an international activist network working to both visibilize and organize against racism [24]. At the same time, attacks against any BLM action from both police and anti-BLM counter protestors (i.e., militia) intensified as well [25,26].

Efforts to maintain racism and protect the national patriotic narrative, however, also intensified. Like the growth of the BLM movement, key events were instrumental in heightening the racial fear that conservatives held. In the summer of 2019, *The New York Times Magazine,* released a special issue called the "1619 Project", which reframed U.S. history to visibilize how the institution of slavery, enslaved Africans, and their free descendants were central to the American narrative and nation-state building. Led by journalist Nikole Hannah-Jones, the 1619 Project immediately took off at a national level and now includes a book, podcast series, television series, section of the New York Times, and numerous other teacher development learning opportunities and curricular materials. The initial positive attention was so great that it took notice from conservative historians, pundits, and the government during the shelter-in-place orders. As a response, to seize back control of the U.S. narrative of exceptionalism, the "Saving American History Act" bill was introduced in July 2020 to deny any federal funding to public schools who utilized the 1619 Project as curriculum [27]. Hannah-Jones, herself, also felt numerous forms of backlash, including being initially denied tenure from the University of North Carolina at Chapel Hill's Board of Trustees despite recommendations from the department she chaired, Race and Investigative Journalism [28].

Around the same time, conservative education and anti-CRT proponent Christopher Rufo first stumbled upon critical race theory and saw the possibilities of manufacturing a scapegoat. Through right-wing tactics of distortion, distraction, and fear mongering, Rufo explicitly sought to redefine CRT and demonize it as a threat to America and Americans on social media, television interviews, blog posts, YouTube video, and his own publication, the *Critical Race Theory Briefing Book* [23,24]. He was intentional and made no attempt to hide his strategy to manufacture a new and clear villain by redefining it as an existential and weaponized threat to Americans [23,24].

His work quickly caught the eyes of high-profile conservatives, most notably, Fox News host Tucker Carson, and the 45th President of the U.S ("the 45th"). The 45th, in particular, worked directly with Rufo to craft a ban on any federal agencies using "tax-payer dollars" to conduct anti-racism and Diversity Equity Inclusion (DEI) training [24]. Following this move, the 45th launched the "1776 Commission", which worked to put forth a "patriotic" narrative that downplayed racism and inequality and emphasized a unity predicated on seeing slavery, segregation, and ongoing racial injustice as aberrations in a fundamentally just and exceptionally free nation [27] (p. xxvii). The avalanche of assaults continued at an immense pace in federal and state policy, social media, news outlets, school districts, local activism (e.g., parents, unions, etc.), and the public sphere in general. The

once "nice" form of racism as microaggression and "colorblindness" transitioned back to its more violent forms [29]. Through the continued distortions, distractions, and fear mongering that proliferated, CRT became the parent to an entire family of villains (even if there was no relation at all) which included Marxism, ethnic studies, Diversity Equity and Inclusion, culturally responsive teaching, culturally relevant pedagogy, critical Pedagogy, anti-racist education, social-emotional learning, restorative justice [24], and more recently, "wokeness", affirmative action, and even university students holding multi-day demonstrations (including encampments) to protest the Israeli militarized occupation and violence to Palestinians in Gaza. One way to maintain watchdogs on educational institutions who might have been engaging any of these disciplines and actions was through the campus-watch formula created by right-wing activist and writer David Horowitz [23]. This formula, in particular, was how I started to enter the radar of conservatives at the national level.

I am no stranger to racist vilification, but my most recent experiences at the national level have helped me to develop more precision in my analysis of racism, whiteness, anti-Blackness, and white supremacy. My most recent experience, in particular, was from conservative reactions to a published article of mine entitled, "A Love Letter to Educational Leaders of Color: CREWing UP with Critical Whiteness Studies" [6]. As I was first writing that article in the summer of 2021, I was simultaneously experiencing direct racist attacks initiated by a campus-watch publication called the College Fix. As a professor of educational leadership, I wrote that article during the global pandemic and racial reckoning that were simultaneously occurring to remind those engaged with challenging racism within their work contexts, especially my doctoral and administrative credential students, that the resulting aggressive, right-wing reactions that follow are emotional, predictable, and formulaic. Within that article—a critical race love letter—I humanized the experience of dangerous work while also engaging critical whiteness studies as a theoretical framework to analyze racial discourse that sought to (re)assert subjugation and (re)enforce white supremacy.

Since being published, that critical race love letter has been used as curriculum in multiple spaces, including my university, where I help to co-facilitate a year-long, provost-sponsored faculty fellowship called the *Anti-Racist Liberatory Pedagogy Academy (ARLPA)* [30]. As a community of praxis, our learning experiences are grounded in CRT and critical pedagogy [2]. Among the readings that the fellows are required to read, discuss, and apply is my Love Letter to Educational Leaders of Color. Just before the 2022 fellowship academy launched, my article and work with ARLPA became the center of a news story published online by the right-wing, conservative publication Breitbart News. Within the first 48 h of releasing the news story, approximately 1500 vitriolic comments on their website and dozens of emails directly sent to me used violently racist hate speech.

The nature of the Breitbart article and resulting comments and messages included pillorying language that was scornful, demeaning, and threatened violence which are the typically expected "not-so-nice" behaviors resulting from commonsense notions of "racism". This contrasted with "nice" enactments of racism that include white-privilege ignorance and colorblind ideology that hide behind claims of "I didn't know", "that wasn't my intention", or "you're taking me wrong". Our more current "post-colorblindness" moment [31], (Leonardo & Dixon-Román, 2020), however, returns to more pronounced enactments of "not-so-nice" racism. In colorblindness, race and racism were outright rejected or minimized while defending white advantages through coded and implied language and practices [6,32]. (Doane, 2020; Reyes, 2022). Because of our ever-changing sociopolitical climate where the tactics of racism morph and evolve, post-colorblindness allows for whiteness to reposition itself to permit white people to consider themselves as just another minoritized and marginalized group that suffers from racism [31]. With this in mind, the Breitbart article and resulting comments and messages also included accusations of *me* being "divisive", "hateful", or "racist" (more unique to our changing political and social climate). My critique of whiteness and white supremacy was read as *me* being racist. Put another way, I was not being nice.

In spite of the assaults, like many institutions, my university's executive leadership did not provide much support other than alerting campus police about possible physical threats and our information technology department so as to put some filters on my incoming email. I did not even receive any communication from executive leadership to check in with how I was doing or what *I* needed. Because of my consistent presence and engagement with my campus' faculty and students, I did, however, receive great support in the form of advocacy, public declarations as well as written expressions of solidarity with my racial justice work and me. Graduate students and alumni wrote a collective letter of advocacy directly to our university president. Our faculty senate president openly expressed senate leadership support for me at the beginning-of-the-year opening convocation that included executive leadership, college leadership, faculty, and staff. Deans and department chairs had expressed solidarity in similar public as well as written settings. The following was a written response from my college dean to the students and alumni who advocated for me, "I am writing to assert that I stand with G. Reyes. I stand behind his scholarship, and I will openly defend him. I stand with faculty and students whose scholarship challenges hegemony" (personal communication, 3 August 2022).

The lack of institutional support could easily be interpreted as maintaining the optics of having a neutral stance. However, this excerpt from the graduate students and alumni provide a more critical analysis.

> When our university stays silent, it signals to groups such as Breitbart that it is acceptable to express hate and to use violence. Local politicians and school board members will take notice and repeat patterns of silence. This sends the message at the local level that it is safe to express hate and violence and to put our children and families at risk. This action letter is designed to ensure that CSUEB demonstrates its anti-racist stance and provides clear and transparent language and actions that support anti-racist and anti-Blackness work that students, staff, and faculty engage in to dismantle systemic racism. (personal communication, 23 July 2022).

I suppose I should be "lucky" to have not been blamed by my institution for my receipt of racist aggression. However, as the advocacy letter excerpt brilliantly illuminated, lack of institutional support in regard to defending racial justice within real-world cases of racism is merely another manifestation of protecting white supremacy over protecting individuals such as myself in racial justice work. In spite of how devalued and unsupported I felt from my university's response, lack of institutional support, however, does not necessarily equate to lack of cultural and collective support, so it deserves to be said that racial justice work is most effective when it is collectively done in transformative solidarity. The outpouring of transformative solidarity and radical, decolonial, and humanizing love that I received from colleagues and students was affirming to me and told me that I had been doing racial justice work with integrity.

Given this, the "not-so-niceness" of this current critical race love letter is one that works to continue my praxis-guided participation in cultivating transformative solidarity. In doing so, this current letter (i.e., "study") directly confronts with specificity the racist discourse that resulted from Breitbart. Methodologically, I use those messages as the data sources of analysis. I do not perform this analysis for the sake of analysis, but as a way to demonstrate how to respond to racist discourse and engage in a praxis of love as a way to heal. In essence, this love letter offers a critical race *clap back* as a method towards self-preservation.

To love ourselves, particularly within a country that does not love us back and would rather us hate ourselves, cannot only be *dangerous* to those who are afraid of us asserting our humanity, critiquing and healing from intersecting systems of oppression, and working towards our liberation. To love ourselves must also be *hopeful* and *aspirational* to those of us living in a country that does not love us back and would rather us hate ourselves. In other words, this letter cannot just be "anti"—it must also be "pro". Loving ourselves must *also* be a communal and political endeavor to address the present while working

towards self-preservation. The necessity to self-preserve is rooted in a sociohistoric analysis that identifies intersecting systems of oppression as impacting minoritized communities in unhealthy and violent ways more severely than dominant groups. Influenced by this intention, self-preservation practices include those that are joyful, loving, healing, grounding, growth-oriented, caretaking, communal, reflective, mindful, and consciousness-raising that help maintain our humanity and health within a society that operates under conditions that create harm, unwellness, dis-ease, inequity, exploitation, and injustice towards minoritized peoples.

As self-preservation requires love, this critical race love letter has no choice but to be a love letter. It is a love letter particularly to those of us who know firsthand what it means to experience racial harm over and over again, which Matias [14] argues is a repetitive cycle of abuse intended to maintain whiteness. How then do we interrupt this cycle of abuse? First, we must deeply understand how the cycle of abuse functions. To do that, I now make transparent the critical lenses (i.e., "conceptual framework") that I apply to analyze and clap back against the racist discourse from the Breitbart article as well as comments and messages that followed (i.e., "data").

## 3. Conceptual Framework: White Emotionalities and Whitelashing

To love ourselves as Black, Indigenous, and other People of Color, particularly within a country that does not love us back and would rather us hate ourselves, requires us to understand that the behaviors that result from not only movement towards racial justice but even discussions about race are only surface-level, emotional responses [33]. Undergirding these emotions are what Matias [33] calls *white emotionalities*. She elaborates, "the emotionalities of whiteness are the core human emotions and are root causes for manifestations of defense mechanisms; emotional defense mechanisms that surface up in ways that seek to protect and keep those deep-seated core values hidden" (p. 3).

Given this articulation from Matias, defense mechanisms are viscerally experienced as outward and observable behaviors. The influencing emotions, however, are internally felt and not immediately observed. In other words, how we feel influences how we act. This dynamic is very typical of being human. White emotionalities, however, add an underlying ideological layer and power dynamic to this system of human emotions. Matias [13] specifies "whiteness as a social power and ideology normalizes white emotionality as nonracial" (p. 26). What we believe and what we value, whether we are aware of it or not, influences what we feel. Often times, people are lauded for professing their "truth". "Feelings", particularly from those who hold more power, are sentimentalized and tend to be left untouched. However, rarely do we ask ourselves and others, why do we feel the way we feel in the first place, and what is really underneath this feeling?

Matias [13] continues to assist us with examining these layers of feelings with an understanding of emotional *diminutives*, which aim to perform a superficial level of positive emotions such as love and care in the form of "niceness" but mask other emotions that are not as socially acceptable, such as disgust. Anthropological linguist Maïa Ponsonnet [34] adds "diminutives are used to attenuate the effect of a speech-act and manage 'politeness'" (p. 8). As such, the deeper human emotions that might be felt when white emotionality is activated include resentment, rage, disgust, contempt, shame, fear, and guilt. These emotions would then manifest outwardly as defensive behaviors, which include deflection, distortion, denial, rejection, avoidance/withdrawal, aggression, and crying. Similarly, attitudes and dispositions such as niceness, politeness, and even professionalism are both emotionally and ideologically laden. When these emotions, defense mechanisms, and attitudes are activated through white emotionality, they defend, protect, and reinforce whiteness particularly when it is perceived to be under threat. The defense and protection of whiteness, which is a socially constructed and normalized set of beliefs, practices, discourses, and values, then work to reinforce the condition of white supremacy [6]. This entire architecture comprises a system, which means that it functions as an interconnected set of mechanics that are repeatable and therefore predictable regardless of time and place.

Bonilla-Silva [35] refers to this specific architecture as a "racialized social system", (p. 467) which essentially is where a racial hierarchy determines the distribution of power.

As white supremacy can often be misunderstood to only mean white masks, cloaks, and (tiki) torches used to instill fear, subjugate, and annihilate non-white people, its use of the processes of racism to protect whiteness actually operates in more sophisticated and nuanced ways with very particular functions and purposes. Matias [13] help to understand this process of defending whiteness by asserting "resentment gets emotionally expressed to (a) further subjugate people of color; (b) reappropriate the language of racial justice to justify subjugation, and (c) emotionally recenter the needs of Whiteness" (p. 11). As behavior, this resentment is named by Carter et al. [29] as *whitelashing.* Specifically, whitelash refers to "individual, institutional, and/or structural countermeasures against the dismantling of white supremacy (the racial status quo) or actions, real or imagined, that seek to remedy existing racial inequities" [15] (p. 258).

Overt cases of whitelashing, for example, can appear as physical aggression through the mobbing on 6 January 2021 at the U.S. Capitol [15]. On the surface such a display of white nationalism and white supremacy was encouraged by the 45th to provoke an insurrection against the results of the 2020 election that he lost. On a deeper level, as difficult as it was for many to witness, the whitelash was predictable. It was predictable as an embodied and discursive reaction, as well as a set of demonstrated emotional and psychosocial behaviors that feared racial progress within a context that is designed to sustain structural white supremacy.

Given this discussion on white emotionality and whitelashing, it is my hope that this exploration and demonstration can help you to imagine how you might use such critical lenses to not only notice but respond back against racialized discourse that might come your way. Before engaging these critical lenses to analyze the racist messages (i.e., "data") I received, let me first make transparent how Critical Race Hermeneutics guided my approach (i.e., "methodology") to collect and analyze my data.

## 4. Methodology: Critical Race Hermeneutics and PRAXISioner Tools

### 4.1. Critical Race Hermeneutics

Though it may seem odd to discuss a "methodology" in a love letter, think of it this way. I wish to make transparent my thinking in *how* I approached what I learned (i.e., "findings") from this particular work (i.e., this "study"). In essence, in this part of my letter, I share how I did what I did—in particular, the inquiry, interpretive, and sensemaking process I went through from being the subject of the Breitbart article and reading the online and emailed comments. Given that, I sought to explore this question: *Within a white supremacist national context, how do I interpret and make meaning of racial texts about and directed towards my work and me?*

As communication is not only an expressive process but also an interpretive one that is central to the human experience, discourse is fundamentally hermeneutical [12]. In considering this along with CRT's recognition of the intercentricity of race and racism in the U.S., it was a clear choice for me to use Critical Race Hermeneutics (CRH) to guide how I interpret the presence of white supremacy in the racial texts I received. Through CRH, the presuppositions that guide the interpretive process are inherently made transparent, which is that white supremacy exists in the U.S in a normal*ized* (i.e., made to feel normal), natural*ized* (i.e., made to seem natural), and rational*ized* (i.e., made to seem justified and sensible) way. Allen [12] elaborates:

> In short, CRH is a study of how communication is distorted by a white supremacist social structure, turning discursive exchanges into everyday forms of racialized material, psychic, and symbolic violence. It seeks to show how language and communication is a site of conflict and domination, a place where white supremacy not only operates ideologically but also where the structure of white supremacy is, itself, reproduced. CRH works to interpret, more so, reveal the unconscious of the objective reality of white supremacy in subjective forms. (p. 18)

Allen argues that white supremacy is an objective reality (i.e., it materially exists) that is experienced in subjective ways (i.e., real and nuanced human experiences). Similarly, antiblackness (one word, lowercase b) is an objective reality that is systemically designed to impact both Black and non-Black people. Anti-Blackness (hyphenated, uppercase B) represents the subjective ways Black and non-Black people experience the application of antiblackness as system [36].

Because of the subjectivity of white supremacy, it may be helpful to think of the distortion that Allen speaks of as akin to the expression "keep it 100", which is similar to the older saying "keep it real". The use of these expressions tend to be applied when someone is truth telling. However, the converse can also apply in this sense—what happens when our conception of reality is distorted by the lens, practice, and structure of white supremacy? How does that impact our "truth telling?" In short, our conceptions of reality invisibilizes how our "truth" is fundamentally based on relations of power within a white supremacist context. In other words, our "truth" is distorted.

Given this, I am compelled to make transparent the following intention. In spite of my elaboration on white supremacy and how CRH works to visibilize racially normative sensemaking within a racial hierarchy and discourses directed by whiteness and antiblackness, I do not make any attempt to "prove" that white supremacy, whiteness, antiblackness, anti-Blackness, or racism exists. It is my *presupposition*, as both CRT and CRH makes clear, that white supremacy, whiteness, antiblackness, anti-Blackness, and racism exists and are real, normative parts of the U.S.'s foundation. As such, through CRH, I look for the specific presence of white supremacy, whiteness, antiblackness, anti-Blackness, or racism within the texts that I source as my "data".

*4.2. The PRAXISioner Tools*

As this critical race love letter uses CRH to inform me, as a researcher, to move into, through, and out of the research context, I use the concept of a "*PRAXISioner*" identity to guide my overall purpose, which is to be of concrete help to my local and global community in our struggle for community preservation and liberation. The PRAXISioner identity is rooted in Freire's [2] conception of praxis, which involves engaging the language of critique to problem-pose one's material conditions within a cycle that includes engagement with critical knowledge/theory, self-reflection, dialogue, and action. This cycle is a communal practice, which means that the PRAXISioner works while being embedded in and with marginalized communities towards their self-determination, community actualization, and cultural perpetuity. In doing this, they understand that their context is complex, fluid, and requires multiple frameworks to support the critical sense-making process.

Because of this, PRAXISioners put theory into practice and practice into theory in such a way that centers upon narratives of hope and possibility rather than singularly defined narratives of pain and pathology. As practice, the PRAXISioner is concerned with sharpening their analysis through deepening their ongoing learning and self-reflection of critical theories, "street wisdoms", and history. As critical, humanizing, culturally sustaining, and decolonial pedagogues and researchers, they approach healing as a process. Liberation is unobtainable without intentionality towards healing as an ongoing process rather than as a destination. PRAXISioners work towards a futurity absent of systems of oppression, while working in transformative solidarity with those most systematically harmed. In this sense, the PRAXISioner is not only an identity but also a methodology, pedagogy, and practice.

For this love letter, the specific processes used to engage the PRAXISioner as a methodology, pedagogy, and practice are codified as such: (1) problematizing, (2) visibilizing, (3) reframing, and (4) reimagining. As processes, each can occur within any of the praxis phases. For instance, problematizing can occur during the knowledge, reflection, dialogue, and/or action phases. The same is true for any of the other PRAXISioner processes to be layered with any of the praxis cycle phases. Similarly, multiple PRAXISioner processes can

occur within a single praxis cycle phase. In essence the PRAXISioner tools are layered in non-linear ways with the praxis cycle as shown in Figure 1.

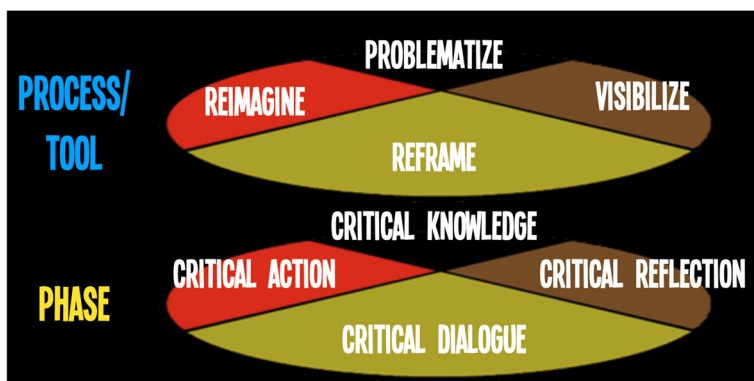

**Figure 1.** PRAXISioner tools layered with praxis phases.

Problematizing requires a critical inquiry into and analysis of situations, systems, and conditions. As a process, problematizing asks the questions: Why are things the way they are in the first place? How has this situation influenced my story, my behaviors, my thinking, and my conception of self and others? Rather than pathologizing people in thinking they are the problem by asserting, "they can't, they won't, they don't, they never will", problematizing gets below the surface level event in order to get to the invisibilized root beliefs, values, and historical patterns that underlie the situation, system, and condition [22].

The process of visibilizing extends the analysis begun during problematizing. Embedded in the process of visibilizing is a recognition that an ideology, core belief, value, and/or pattern in history has been systematically invisibilized [22]. Visibilizing allows PRAXISioners to draw attention to and precisely name the normalized, naturalized, and rationalized system at work. In doing so, the process of visibilizing disrupts the intentions to make a particular situation, system, or condition feel normal, seem natural, and be justified in its explanation. Critical questions to ask include: How do we break down this situation into its moving parts? What critical analyses can shed light on this situation? How is the problem named?

Visibilizing is a rupturing process. Once a rupture begins to occur and an underlying problem can be visibilized, something different can emerge. The situation, system, or condition can be reframed. Reframing engages frameworks and language that liberates and transforms from the status quo to open up new opportunities. Reframing helps us to see, think, and do things differently than the dominating and oppressive norm. It allows for a (re)creation of processes and practices that create space to disrupt traditional power relations and move towards radical possibilities [22]. Critical questions to ask include: How might we be able shift the gaze to think about this situation differently (from the dominant)? Which logics need to be made transparent to reveal the intentionality of this reframing? Which different language can be used to help in shifting the previous thinking to this reframed way?

Reimagining is a communal endeavor that works towards a futurity absent of the oppressive systems that maintain relations of domination and subordination while reproducing harm and sustaining woundedness [22]. As a process, reimagining catalyzes a radical intentionality and healing-centered mindfulness in the present that requires us to build from our work in problematizing, visibilizing, and reframing in order to redesign ways to be and become *beyond* the current construct. Questions to be asked include, given the analysis and reframed way to think about this situation: What might we imagine how things could be? How do we revise our actions and behaviors to reflect the world we want and deserve to live in? How do we adjust and refine what we learn along the way?

### 4.3. Towards the Clap Back

I have been asked many times how I respond to the racist messages I receive, particularly those sent electronically—*do you reply to those emails or post comments on those blogs/social media posts*? I do not. Here is why. The messages are consistently emotionally irrational in nature, so I do not believe that there is any response I could make that would fulfill any productive objective. I do not believe anyone who would send me messages (actual messages received) that say, "Dirty fucking n__ger" or "What kind of left-wing, cock filth RACIST pile of shit are you?" or "Suck a dick you retarted racist fuck moron" would be able to openly receive any responses that were remotely critical or logical from me.

Rather than send individual, direct replies, I have made my responses public as a service rooted in communal education, healing, and affirmation to those who have had or know of others who have had similar experiences, see [37]. My intentions with my public responses have aimed to problematize the experience, visibilize and name the oppressive system at work, reframe how to make a meaningful response, and provide a reimagined way to respond. Some responses have been through published scholarship, such as my Love Letter to Educational Leaders of Color [6]. Some have been through art campaigns, such as with the Cross This Out movement [37]. Some responses have been pedagogical through public presentations, professional learning, and critical research classes I have taught (please email me for more curricular information, such as with my work on critical race poetry or using racist discourse as a data analysis coding and meaning making lesson). In doing so, I believe I have been able to mobilize communities to engage praxis in concrete ways within their own contexts. As such, I wish to do that again within this critical race love letter.

Through the conceptual framework and methodology shared earlier, I collected hundreds of comments and dozens of emails and presented my analysis of them in the form of a spoken word poem entitled "Dear haters". The poem is based on a prompt that I have used in my Critical Race Methodology doctoral class, which is as follows: *Think of a time where someone or a group of people said and/or did something to you that represented an act of systemic oppression. What would you say back to them? How would you speak truth and lace them up with game*?

The poem in itself is the "data analysis" and unpacks the racist system at work in relation to the direct quotes I used from some of the messages received. I intentionally allow the poem to stand on its own within the *Data Analysis and Discussion* section that follows. As data analysis, there will be apparent areas where you will notice how I *problematized* the racist situation and *visibilized* the system at work. As you finish reading the poem, I encourage you to reflect upon how the poem in itself reframes both the way we think about what data analysis and knowledge production might look like as well as how to respond to racist discourse in a public way. Additionally, I invite you to *reimagine* how your students, teachers, and you could respond to injustice in meaningful, creative, and performative ways. To be clear, when I highlight *performative*, I do not refer to a superficial, image-fabricating act that is ingenuine. No. I refer to performative as a descriptor of *performance* as a site of struggle. I refer to how the power of performance could be engaged to mobilize the masses and challenge the performances of power [38,39].

As you read the poem and come across the direct quotes, please prepare yourself for possibly being activated. If you do find yourself activated, please remember your self-preservation practices and that you are not presently in the space that caused the harm of which you are reminded. You have likely been on a healing journey from that/those space/s and are a much stronger person since then. And if anything, please remember that you are reading a love letter that is for you, that is for me, that is for us.

### 4.4. Data Analysis and Discussion: The Critical Race Clap Back

**Dear haters,**
You ask me why
I am being

"Divisive"
"Hateful"
"Racist"

So I think
And think
Very seriously

And consider
Saying the truth
And have it said like this

You say my work to engage race-consciousness as an analytic
Is "sick" and "divisive"
You "don't see color"
You say
Yet you decisively distort and
Devise an alternative history erasing 1619 and Liberated Ethnic Studies
to suit your comfort today
Because "look how things have changed" you say
Oh and "a guy like me was even able to get a PhD", and by the way
When you say I "should be grateful each day"
You assume I'm not,
So you say I "obsess over my betters"
That my "critical race love letter
Was out of touch with reality"
That "there's nothing loving" about me and
"The backlash against me will be what I deserve and need"
You say I'm "going to fall"
But you the one who's falling
Look at you all in your feels
Stumbling over your back heel
You afraid folks of Color like me are gon' steal
your privilege
your entitlement
so you clench tighter
face getting ghostly whiter
And you're losing your grasp
While your privilege crumbles through your fingers
and that makes you emotional
scared
so you whitelash out
As your light flashes out
Eyes wide blinded
swinging arms recklessly as your entitlements get smashed out
And yet you call me "divisive"
because I critically discuss race?
But check it out homie,
Who institutionalized division in the first place?

But I still see you there
Asking
No
Telling me why
I am being

"Divisive"
"Hateful"
"Racist"

So I do think
Very seriously

About saying the truth
like this

You deflect when you say I'm "playing the victim"
And call me a "retarded racist fuck moron"
You say I "feed on
Racism for coin"
Your bros join on
Saying that "those who can't do, teach"
and my "degree from Cal
is from a gumball machine"
But your one syllable words that try to demean me
like a child calling somebody something mean see
is emotional & infantile
All the while you don't even know what you mean
You clearly don't know that white people are not the same as white supremacy–
a system to oppress by normalizing whiteness
and making seductive what it means to be and become
Employing the process of racism to shun that which does not fit
It pushes it
over the side
down the cliff
tumbling
to be buried in rubble and
never to be seen again
to be remembered
And you say that I'm "in a bubble?"
That I'm "out of touch with reality"
Look son, I been had your number
keep telling yourself that bedtime story
you can't get rid of us that easy
We fight racism because we have to
Because as the homie Cam says, lives depend on it
And since you say I'm "just a teacher"
Sit down so I can teach you
Racism is a system of societal power
for a group of people of a certain race to dominate
Expecting other races to cower
while they tower over
Oversee
But when you feel that power is threatened
your whitelash explodes into emotional debris
Tiny pieces of rage, guilt, defensiveness, disgust, tears
falling from sea to shining sea
Polluting the water
Distorting your understanding
All while maintaining the idea and perception
that you are superior to us.

But I still see you there
Glowering
All caps
Exclamation points
Screaming "YOU COCKSUCKING HOMO BABOON"
Throwing more N words than a Tarantino film
Asking me why I "hate being Black"
Vilifying me when you say, "You're a coon. Now I understand.
FUCK YOU, YOU BANANA SUCKING AFRICAN JUNGLE DWELLING BOONDOG
ASSHOLE"
Because of course you feel entitled to assign race
To define race
To distort race
To assign Blackness as punishment
Antiblackness as both whip and chain
Cuz that's how white supremacy works
And I see you comin.
So wait a minute
Shh
Hear that
Your tantrum
All that
Was just you reinforcing white supremacy.

So clearly I do think
And think
Very seriously

And consider
Saying all this truth
But simply say

Yeah, I know you're emotional
afraid
You should be
But my critical race letter wasn't really for you anyway
It was for us
And I write it out of love for us
So Ready or Not
Here we come
You can't hide
We've already found you.
And that got you shook huh?

## 5. Conclusions: Embracing the Not Niceness of Dangerous Work

Have you been maintaining your breath? If you are feeling that your breath has been erratic or short, please take a moment now to attend to your body and its tension. Please also remind yourself that you are loved and worthy of love. Your ancestors are no doubt proud of you and who you are becoming. When you are ready, process your emotions and thoughts. If you do, as you know, it helps to reflect with community—to engage in collective care. Your presence on this Earth is important and necessary—even if some might find you and your work dangerous—so know that the relationship you are developing with this love letter is aimed to support your continued journey of wellness and transformation.

I say all this to you to model that engaging with racial hostility and violence requires active practices for self-preservation. Self-preservation in itself, particularly for those of us

who are educational leaders working to reimagine schools and higher education, can be seen as dangerous, because we are not supposed to survive, let alone thrive, within our current racialized social system. Self-preservation is a revolutionary act. Thus, we must remember, for the sake of our community-preservation, that sometimes, when we have a traumatic experience or get reminded of such experiences, we do not realize how much we embody and internalize our reactions. When we do not release those energies, they can get stuck and manifest in unhealthy ways. When we do not process such experiences with critical lenses, we might face gaslighting from those inflicting the harm—minimizing, deflecting, or outright dismissing how we experienced the situation. We can even get stuck in our heads replaying the experience—thinking about how we should have responded or asking ourselves, "did that just happen" or "maybe I'm being too sensitive" or "maybe I shouldn't have said anything". Over time, the accumulation and compounding of our dis-ease can turn into disease.

If we find ourselves experiencing those reactions of dis-ease, we must also take notice and pause, so that we could remind ourselves that we must contextualize those responses within a vast, insidious, racialized social system that not only determines but reinforces the distribution of power. Within a system, there are designed and therefore predictable outputs. Therefore, someone who benefits and wishes (whether knowingly or not) to continue benefitting from racism and white supremacy will *predictably* demonstrate defensiveness. This is part of how the system functions. Our experiences with racial hostility and violence are not unique, individual examples of "hate" but rather, like the racist discourse resulting from the Breitbart article, all part of a well-functioning system.

Thus, when we return the question of *nice for whom*, we have to ask ourselves: Who might niceness coddle? What does niceness have to do with combatting and countering racial injustice in the first place? At a very nuanced level, we also have to ask: How is niceness being weaponized to both pacify racial justice work and suit the needs of whiteness and antiblackness? In other words, niceness is one of the many tools to suit a win–lose situation. Niceness can be used to defend behaviors stemming from whiteness—"*I didn't mean it that way*", "*I was just joking*", or "*I didn't know*". Niceness can also be used to accuse others who challenge whiteness—"*Why are you taking this so seriously*", "*Why are you so angry*", or "*You're being divisive*".

I finish this critical race love letter on the same week as Super Tuesday, where the 45th has emerged as the frontrunning Republican candidate for the 47th President of the U.S race. I finish this critical race love letter on the same week where talks for a Gaza ceasefire are still at a standstill. I finish this critical race love letter on the same week where a prominent critical whiteness homie of mine was removed by a school board as the keynote speaker for a district event where she was invited over a year ago for being too "radical". I finish this critical race love letter a few weeks after the founder of Woke Kindergarten (WK), LLC received a scrutinizing and invasive letter [40] from Bill Cassidy (R-LA), ranking member on the U.S. Senate Committee on Health, Education, Labor, and Pensions, expressing extreme concern over the adoption of the WK program in a Bay Area elementary school.

I say all this because our work as PRAXISioners is far from over. At times, we might feel all is hopeless and wonder what the purpose of our fight is. We might feel alone. I assure you, we are not alone. You are not alone. The fight is for our collective humanity. The fight is for the seven generations that follow us. Allow your training, experience, and knowledge to empower you in your continued fight. Rely on the tools you have gained, but also continue to hone those tools and expand your toolset. Continue to build in transformative solidarity with your communities. We cannot afford to play defense with the ever-morphing rules of niceness, so let us embrace our collective not-so-nicess. Let us embrace our collective dangerousness. Let us remain focused in our collective liberation. And let us do so while grounded in love.

With love,
G

**Funding:** This research received no external funding.

**Data Availability Statement:** The raw data supporting the conclusions of this article will be made available by the authors on request.

**Conflicts of Interest:** The author declares no conflict of interest.

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
