# Peer review of "Nice for Whom? A Dangerous, Not-So-Nice, Critical Race Love Letter"

_education, doi:10.3390/educsci14050508_

Round 1

Reviewer 1 Report

Comments and Suggestions for Authors

This is an important piece and welcomed invitation (love letter) at a particular moment in time when critical scholars and our student colleagues may be in need of this kind of psychological balm and navigational toolset. I also appreciated the author's grounded praxis of "returning to the source," as Amilcar Cabral might put it, to both offer medicine to the people (the source) and to address the harm that inspired the current piece (another source). I am left wondering exactly when the description of the Critical Race Love Letter ends and the actual epistolary offering itself begins. This need not be explicitly formatted with marked distinctions within the piece. Instead, the author might consider providing more clarity in terms of a roadmap earlier in the writing.  

Author Response

Peace and greetings, Reviewer 1.

Thank you so much for your affirmations and suggestion for clearer delineation as to "where the Critical Race Love Letter ends and the actual epistolary offering itself begins." I believe that the Critical Race Love Letter and epistolary are one in the same. As such, I made some slight revisions to make this belief more transparent within my article. 

With gratitude and respect,

G Reyes

Reviewer 2 Report

Comments and Suggestions for Authors

The author writes in the form of a critical race love letter to BIPOC educational leaders engaged in anti-racist work, analyzing the discourse of racist messages as a response to an earlier published article with the aim of ensuring readers feel less isolated and not alone in experiencing racist vitriol in response to anti-racist work. The author uses “niceness” (cf. Castagno) and interrelated CWS concepts as an analytic lens to help readers/letter audience, "imagine how you might use such critical lenses to not only notice, but respond back against racialized discourse that might come your way.” The author goes on to articulate their purpose: "I do not perform this analysis for the sake of analysis, but as a way to demonstrate how to respond to racist discourse and engage in a praxis of love as a way to heal” (p. 6).

I think this piece is very well done and achieves its stated aims. It offers readers many useful/critical concepts (niceness, white emotionality, whitewashing) and an important/mode of analysis for emotionally processing experiences of racist harms in response to anti-racist work, and as a means to contextualize and understand their experiences within a broader and longer context that de-isolates and de-individualizes their experiences and that also lovingly nurtures educators’ commitments, capacity for, and enactments of anti-racism. 

In considering what my feedback contribution might be to this solid piece, I just have the following food for thought: In reading, I began wondering about all the other kinds of emotional and mental fallout people experience and are forced to respond to for engaging in anti-racist work in education. There are many forms of emboldened racist responses to anti-racism that educational leaders of color experience in K12 and higher education that have intensified material consequences  — from doxxing and physical safety threats, to suspension or firing (all of these have happened within the last two years to various colleagues at my own institution, for which we had to organize collectively to respond to, often without satisfactory outcomes from our university leaders). Often, while folks are emotionally processing messages, they are dealing with the ways in which their institutions utterly fail to protect them (and at worst, assist in the perpetuation by blaming faculty/teachers experiencing the racist and other harms! — which makes me wonder how the author felt supported or not by their institution during their experience?) This kind of institutional complicity is happening most viciously right now around student-led protests for Gaza and Palestine, as universities respond with militarized strength and post snipers atop campus buildings (as a current example, likely not at the time of the author writing, but many past instances of campus protests in movements for Black lives, student labor protests, Occupy, etc. have faced similar campus militarized responses across many universities in the US in the last decade or two). 

In response to these institutional failures, I am thinking of the collectives educators form to defend and respond to this harm (one example is the Faculty First Responders — a group dedicated to collecting and communicating resources and outing the strategies and networks of right wing organizations targeting students and faculty in higher ed). 

So, all this said, I think I would have liked to see some more explicit discussion of this broader context/range of racist and sexist/misogynist, anti-queer/trans, colonialist attacks on educators (e.g., existing literature that similarly grapples with/examines contexts of racist attacks on K12 and higher ed educators) to contextualize/qualify the authors’ study as one important mode of processing/inner work and academic clapback we can undertake to continue to do the work. 

Thank you for your work! And solidarity!

Author Response

Peace & greetings, Reviewer 2. 

Thank you so much for your insightful food for thought. I have taken your comments very seriously as I completely agree with all of them. As such, I did make revisions in particular to make transparent my university's lack of support from executive leadership, immense cultural support and praxis led by colleagues & students, and how I felt along the way. I believe your suggestion was crucial to narrating the complex terrain.

With respect,

G Reyes